# Multi-Scale Indoor Scene Geometry Modeling Algorithm Based on Segmentation Results

**Changfa Wang, Tuo Yao and Qinghua Yang ***

School of Mechatronic Engineering and Automation, Shanghai University, Shanghai 200444, China;
changfa1999@shu.edu.cn (C.W.); yaotuo@shu.edu.cn (T.Y.)
*   Correspondence: yangqinghua@shu.edu.cn

**Abstract:** Due to the numerous objects with regular structures in indoor environments, identifying and modeling the regular objects in scenes aids indoor robots in sensing unknown environments. Typically, point cloud preprocessing can obtain highly complete object segmentation results in scenes which can be utilized as the objects for geometric analysis and modeling, thus ensuring modeling accuracy and speed. However, due to the lack of a complete object model, it is not possible to recognize and model segmented objects through matching methods. To achieve a greater understanding of scene point clouds, this paper proposes a direct geometric modeling algorithm based on segmentation results, which focuses on extracting regular geometries in the scene, rather than objects with geometric details or combinations of multiple primitives. This paper suggests using simpler geometric models to describe the corresponding point cloud data. By fully utilizing the surface structure information of segmented objects, the paper analyzes the types of faces and their relationships to classify regular geometric objects into two categories: planar and curved. Different types of geometric objects are fitted using random sampling consistency algorithms with type classification results as prior knowledge, and segmented results are modeled through a combination of size information associated with directed bounding boxes. For indoor scenes with occlusion and stacking, utilizing a higher-level semantic expression can effectively simplify the scene, complete scene abstraction and structural modeling, and aid indoor robots' understanding and further operation in unknown environments.

**Keywords:** object segmentation; face recognition; oriented bounding box; geometric modeling

## 1. Introduction

With the continuous advancement of point cloud data acquisition technology and processing algorithms, more and more researchers are paying attention to the application and optimization methods of point cloud scene geometric modeling. For example, point cloud scene geometric modeling can be applied in areas such as virtual reality [1], autonomous driving [2], environment detection [3], and robotics [4], providing more efficient and safe experiences and services. In daily work, there are a large number of geometric primitives involved, such as planes, spheres, cylinders, and cones. Many complex objects can also be seen as composed of these geometric primitives, and these primitives have mathematical models. By representing the collected three-dimensional point cloud with basic model parameters, the storage space is greatly reduced, compressing the model. Geometric modeling of three-dimensional point clouds [5] not only enhances the autonomy of industrial robot grasping but also provides more information for the field of 3D reconstruction [6], helping to make the reconstruction results more in line with real scenes. This is particularly important in virtual reality applications as it provides more support for virtual and real-world integration. In addition, geometric modeling techniques are also very important in the field of surveying and mapping. They not only enable automatic surveying and mapping, reducing human workload, but also provide security guarantees in certain dangerous measurement environments.

Solving the problem of automatically identifying geometric primitives, such as planes, spheres, cylinders, and cones, from three-dimensional point clouds is a fundamental problem in robot perception of the environment. Solving this problem can reduce the difficulty of robot perception of the environment and bridge the semantic gap between high-level semantics and low-level visual features. Many existing point cloud registration techniques and point cloud polygon mesh reconstruction techniques can reconstruct the collected 3D information well, but these techniques only reconstruct the surrounding environment or study the topology of objects, without recognizing the objects semantically. Therefore, using segmentation results as input is beneficial for achieving comprehensive scene analysis. Typically, the operating scenes of indoor robots consist of objects with regular structures, and with the gradual application of mathematical models in three-dimensional space, combining point cloud segmentation and geometric analysis can achieve modeling of regular objects, which helps reduce the perception difficulty of robots in unknown environments [7].

The main contributions of this paper are as follows: (1) Introducing multiscale neighborhood search to address the instability of feature value computation at a single scale, enabling accurate determination of planar and curved surface types based on dimensional features and curvature features. (2) Utilizing the normal vector relationship between planes or surfaces as prior knowledge, using the Random Sample Consensus (RANSAC) algorithm to verify and extract parameters of known types of geometric primitives. The entire process does not require a training dataset and can quickly and accurately complete the geometric parsing and modeling of indoor scenes.

## 2. Related Work

In order to understand the scene and reconstruct individual objects or the entire scene using the segmented 3D data, usually only a few simple geometric primitives are needed, such as planes, cylinders, and spheres [8]. However, acquiring complete object models from real scenes can be challenging, making some matching-based methods unsuitable. Therefore, methods that combine object segmentation and geometric modeling have been proposed, mainly categorized into two types: entity-based modeling and surface-based modeling [9].

Entity-based modeling methods directly operate on the segmented objects, identifying geometries and fitting parameters based on extracted features of the entire object [10]. For example, Zhao et al. [11] proposed an approach based on iterative Gaussian mapping, reconstructing geometric objects in indoor scenes based on the distribution of normals on the Gaussian sphere and using improved RANSAC. In addition to traditional algorithms, Li et al. [12] combined deep neural networks and proposed the BAGSFit framework, which uses a fully convolutional neural network to achieve scene instance segmentation. The framework estimates the probabilities of associated geometric types based on the boundary of the entire object, enabling the modeling of three-dimensional primitives with multiple modes. Entity-based modeling methods do not depend on accurate segmentation results but are only suitable for modeling simple shapes.

Surface-based modeling methods, on the other hand, utilize different surface characteristics of geometric primitives to accurately fit the segmented point cloud data with different surfaces. This is followed by operations such as intersection, extension, and merging to obtain complete geometric models [13]. Sun et al. [14] also extract surface features from the scene to identify geometric primitives and construct a graph model based on the color and geometric features of the primitives. They perform graph segmentation to achieve scene segmentation and modeling. Stanescu et al. [15] proposed a method for semantic segmentation and structure modeling of dense point clouds. They utilize an improved RANSAC approach for fitting and refining the geometric primitives, combined with convex hull and support vector machines for classification and merging of the primitives, resulting in structural modeling of indoor scenes. Surface-based modeling methods can reconstruct complex shapes, but the results depend on the accuracy of surface extraction and fitting.

## 3. Methods

Firstly, we perform initial segmentation of the objects based on the superpixel clustering algorithm. After obtaining the complete object segmentation results, the composition of each object's super-voxels and facets, as well as the connection relationships between each component, can be synchronously obtained. Based on this premise, we use the simplest logic to analyze the geometric types of objects, extract the geometric parameters of objects, and model them as prior knowledge. Firstly, according to the covariance eigenvectors, the probabilities of panels belonging to planes and surfaces are calculated, and the types of panels constituting the object surface are initially determined. Then, based on the combination of different planes and surfaces, the geometric models of basic geometric bodies are constructed, and the geometric bodies are divided into two categories: planar geometric bodies and curved surface geometric bodies. In each category, the specific geometric body type is determined based on the relationship between the main matching surface and its adjacent facets. The parameters of the specified type of surface are extracted using the random sample consensus algorithm, and the size of the object is obtained by combining the directed bounding box for modeling.

The entire algorithm process is shown in Figure 1, where the input is the segmentation result of the synthesized desktop scene, and the output is the complete point cloud data generated based on the geometric parameters.

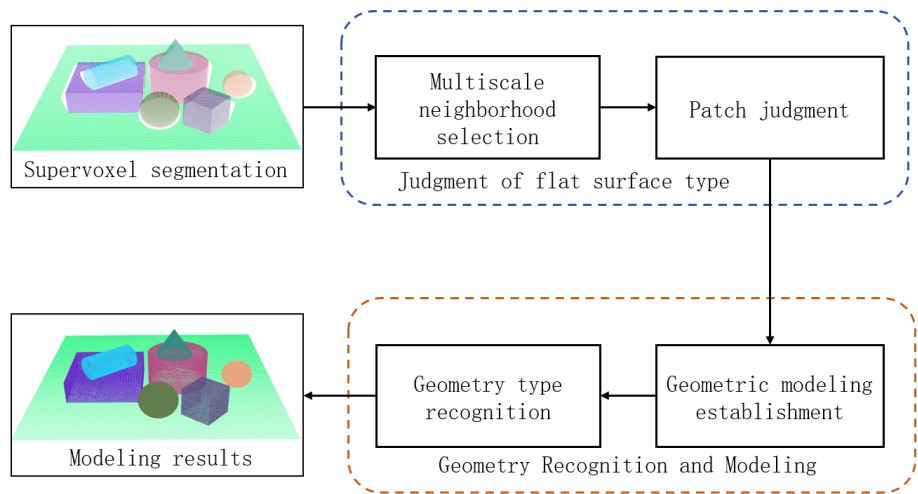

**Figure 1.** Algorithm process.

### 3.1. Determining the Type of Planar or Curved Surface

Common surfaces can be simply classified into two main categories: plane and curved surfaces. Curved surfaces include cylindrical, conical, and spherical surfaces. Different combinations of plane and curved surfaces can form simple geometric shapes. Therefore, before analyzing the specific geometric type of an object, it is helpful to roughly classify its constituent surfaces, which facilitates quick determination of their respective types. This paper focuses on the judgment of planar and curved surfaces, including the selection of neighboring areas and feature calculation based on the covariance matrix.

### 3.1.1. Selection of Search Neighborhood

Currently, the neighborhood of a given three-dimensional point can be divided into a spherical neighborhood, a cylindrical neighborhood, and a fixed-point neighborhood. Spherical neighborhood and cylindrical neighborhood refer to searching for points within the corresponding shape around the given point as neighbors. The neighborhood shape is simple and symmetric. On the other hand, fixed-point neighborhood refers to finding the specified number of points closest to the given point as neighbors, so the obtained neighborhood shape is not fixed. For different point cloud data, neighborhood selection

is generally carried out through empirical or heuristic methods. In addition, considering the three-dimensional structure of the point cloud and the local point density, some single-scale and multi-scale neighborhood search methods have been proposed to meet the demand for accurately extracting geometric features. For single-scale neighborhood search, geometric features calculated using points within a smaller radius range lack stability and are susceptible to noise and outliers. Points within a larger radius range lead to over-smoothing of calculated geometric features, making them unable to reflect the true shape [16].

In this study, in order to quickly and simply determine the type of the current face, while considering point cloud noise and uneven density, the stable characteristics of the central region of the segmented face are fully utilized, and a spherical neighborhood is defined with the center of the face as the center of the sphere. Based on this, a multi-scale spherical neighborhood search is performed to calculate subsequent covariance features, which can accurately and stably determine the type of the face.

The schematic diagram of the multi-scale spherical neighborhood search region for a single face is shown in Figure 2. The red dots in the figure represent the centers of the face, which are used as the centers of the spherical neighborhoods. Three radius values are uniformly selected as search radii, with the minimum width of the face as the upper limit of the radius. The arrows of different colors in the figure indicate the selected radii. By organizing and managing the points within the face using KD trees, the spherical neighborhoods of the current face under different search radii can be quickly obtained for the subsequent calculation of covariance features.

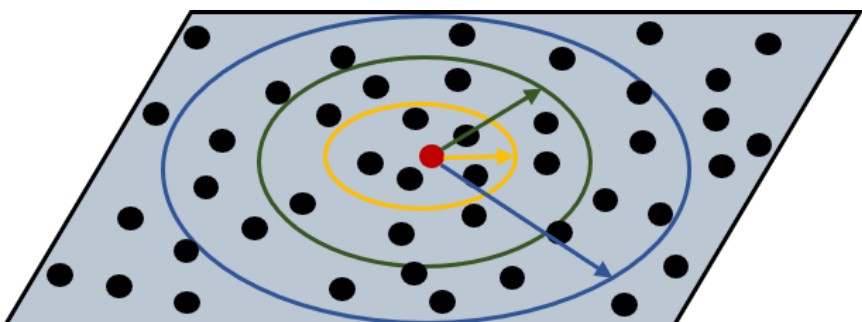

**Figure 2.** Multi-scale search neighborhood plane schematic diagram.

3.1.2. Feature Calculation Based on Covariance Matrix

To construct the covariance matrix using the points within the centroid and its search neighborhood, the eigenvalues $\lambda_1 > \lambda_2 > \lambda_3 \geqslant 0$ of the covariance matrix are calculated through principal component analysis. Based on the different quantity relationships between the eigenvalues, the corresponding dimensional features can be derived. The specific calculation formula is as follows:

$$L_\lambda = \frac{\sqrt{\lambda_1} - \sqrt{\lambda_2}}{\sqrt{\lambda_1}}$$
$$P_\lambda = \frac{\sqrt{\lambda_2} - \sqrt{\lambda_3}}{\sqrt{\lambda_1}} \tag{1}$$
$$S_\lambda = \frac{\sqrt{\lambda_3}}{\sqrt{\lambda_1}},$$

where $L_\lambda$ represents the one-dimensional linear degree, $P_\lambda$ represents the two-dimensional planar degree, and $S_\lambda$ represents the three-dimensional scattering degree, satisfying $L_\lambda + P_\lambda + S_\lambda = 1$. $P_\lambda$ can be used to estimate the similarity between the local shape of the face and a plane. A higher value indicates a smoother face, thus, it can be used to distinguish between planes and curved surfaces [17].

For curved surfaces, taking a sphere as an example, the degree of curvature of the sphere varies with different radius values. A sphere with a larger radius corresponds to a smaller curvature, indicating that the local shape is closer to a plane. The calculation formula for estimating surface curvature using eigenvalues is as follows:

$$C_\lambda = \frac{\sqrt{\lambda_3}}{\sqrt{\lambda_1} + \sqrt{\lambda_2} + \sqrt{\lambda_3}}. \tag{2}$$

When the $C_\lambda$ value is larger, it indicates that the shape of the face is more curved. In order to achieve a more uniform expression, the curvature of the local shape is defined based on $C_\lambda$ as follows:

$$b = (1 - C_\lambda)^2. \tag{3}$$

When the value of $b$ is larger, the current face is closer to being a plane. The probability of the current face belonging to a plane or curved surface is described by combining the planarity $P_\lambda$ and curvature $b$. The calculation formula is as follows:

$$cf = P_\lambda \times b. \tag{4}$$

For a face, the corresponding $cf$ values are calculated based on the points within the spherical neighborhood at different scales as discussed in the previous section, and then they are fused using a weighted approach. The weighted formula is as follows:

$$CF = \sum_{i=1}^{3} w_i \cdot cf_i, \tag{5}$$

where $cf_i$ represents the calculation results at different scales and $w_i$ represents the corresponding weight values. The results of the spherical neighborhood calculation at three scales are weighted using the average noise of the point cloud. The definition formula for $w_i$ is as follows:

$$\begin{cases} w_1 = e^{-t} \\ w_2 = 1 - e^{-t} \\ w_3 = 2(1 - e^{-t}). \end{cases} \tag{6}$$

Among them, $t$ represents the average noise amplitude of the point cloud, which is generally set based on the average density of the point cloud. In the case of high point cloud noise, the value of $w_3$ is larger, which can balance the final value obtained by the larger radius. Conversely, when the point cloud noise is low, the value of $w_1$ is larger, which can use more locally stable features to ensure the accuracy of the results. For a given threshold $CF_{\text{th}}$ for judging planar and curved surfaces, when $CF > CF_{\text{th}}$, it indicates that the current face is classified as a plane; otherwise, it is classified as a curved surface. The proposed method of combining multi-scale neighborhoods with covariance matrix eigenvalues can effectively make preliminary judgments on face types in the presence of noise and outliers in point cloud data [18].

### 3.2. Recognition and Modeling of Regular Geometric Shapes

The surfaces of objects in real indoor scenes are mostly composed of geometric primitives such as planes, cylinders, cones, and spheres. The Random Sampling Consistency Algorithm (RANSAC) can be used to search for the basic geometric primitives mentioned above in 3D point clouds, as well as to extract parameters from specified types of geometric primitives. RANSAC is a hypothesis- and validation-based method that generates hypothesis model parameters based on the minimum number of sample points, and uses all data points to validate and update model parameters. Compared to the process of fitting model parameters using all data points using the least squares method, the model parameters estimated by RANSAC using the minimum subset and local points method are more robust, especially suitable for processing point cloud data with more outliers.

For known geometric primitive types, first the minimum subset required for fitting is determined, such as determining at least three non collinear points on a plane in space. Then, the minimum subset is randomly selected to estimate the parameters of geometric primitives. By determining whether all other data points comply with the current model, the data points are divided into local and external points. Update the model parameters using local points and continue to iterate the above process for the remaining points until the local points are no longer amplified and meet the set threshold requirements [19]. At this point, the optimal parameters of the geometric primitive model are obtained. If the optimal model parameters cannot be obtained in the end, it indicates that the current patch does not match the specified type, thus achieving verification of geometric primitive type judgment.

This paper mainly focuses on the study of simple geometric objects, including cuboids, cylinders, cones, and spheres. By determining the geometric type and extracting parameters of segmented objects, the modeling of regular objects in the original scene is achieved. The surface of a geometric object is composed of geometric primitives, and the Random Sample Consensus algorithm can be used to verify and extract parameters of known geometric primitives, providing a theoretical basis for subsequent geometric modeling. Based on the judgment results of internal faces of various objects mentioned earlier, a basic geometric model graph is established, categorizing geometric objects into planar and curved ones. For different types of geometric objects, their specific types are determined based on the combination information of internal face patches and the relationship of surface mean curvatures, combined with geometric primitive parameters and geometric shape parameters to perform the modeling.

### 3.2.1. Basic Geometric Model Graph

Several geometric primitives can be combined to form some basic geometric shapes, such as rectangular cuboids, cylinders, cones, and spheres, etc. After the initial type judgment of the surfaces composing an object, i.e., determining whether they are planar or curved, the following standard geometric models are defined based on the normal vector relationship of the planar and curved surfaces within the basic geometric shapes, as shown in Figure 3. In the figure, *P* represents a plane, *C* represents a curved surface, $\perp$ represents the perpendicularity between the normal vectors of adjacent surfaces, and the absence of notation indicates that there is no clear relationship between the normal vectors of adjacent surfaces.

Considering the phenomenon of excessive segmentation on object surfaces during the region-growing process, as well as various factors such as single-viewpoint acquisition and varying degrees of occlusion among objects, in order to quickly determine the geometric type of an object, the object can be divided into two main categories: planar geometry and curved geometry based on the combination of surface types within the object. Next, specific situations that appear in different categories will be discussed and processed separately in order to complete the reconstruction of regular objects in the scene.

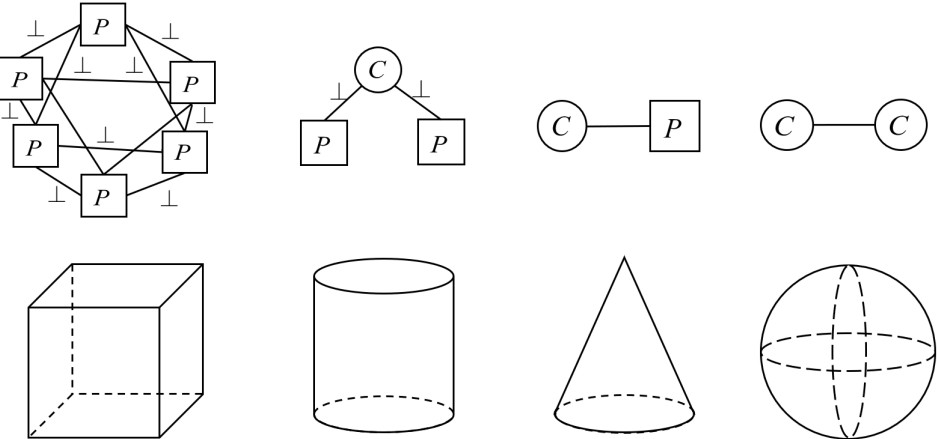

**Figure 3.** Schematic diagram of basic geometric models.

### 3.2.2. Recognition and Modeling of Planar Geometric Objects

The most common indoor scenes are dominated by flat structures such as desktops, floors, and boxes. Using flat surfaces as the main matching surfaces allows for quick recognition of geometric objects that contain flat structures. For objects with only flat surfaces, the largest face is first identified by finding the largest visible face in the current perspective. This face is considered as the main matching surface. The relationship between the face normal vector and the adjacent face normal vectors is then determined. If the normal vectors are perpendicular to each other, the object is considered as a cuboid, corresponding to packaging boxes and other similar structures in the scene. If the normal vectors are parallel to each other, further fusion of the faces is needed, using a larger flat surface as a whole, corresponding to desktops, walls, or floors in the scene.

The initial parameters of a flat surface can be calculated from the center and normal vector of the face. Based on this, the RANSAC algorithm is utilized to fit the parameters of the plane, which speeds up the fitting process of optimal parameters. After determining the specific type of the flat geometric object and the parameters of each face, direct modeling of the geometric object is not feasible without knowing the object's dimensions. To address this problem, the minimum oriented bounding box is computed for the segmented object, according to the construction method of the bounding box. For a cuboid, two perpendicular plane normal vectors can be used as the first and second principal axes, and the third principal axis can be obtained by utilizing the property of mutually orthogonal coordinate axes. Then, the point cloud of the object is projected onto the three directions to obtain the maximum and minimum values of coordinates in each direction. This can determine the length, width, and height of the cuboid, and also obtain the center of the bounding box to determine the object's position in the scene.

Modeling of segmented objects is performed by combining the geometric parameters of the internal planes of flat geometric objects with the dimensions of the oriented bounding boxes. To visually display the modeling results, point cloud data corresponding to the generated geometric objects are obtained using known parameters, representing the reconstruction results. The reconstruction results of flat surfaces and cuboids in the scene are shown in Figure 4. Figure 4a shows the original point cloud data with certain missing parts. The green-bordered box in Figure 4b represents the oriented bounding box for the object. Figure 4c shows the overlaid result of the reconstruction and the original point cloud data, from which we can observe that the reconstruction effectively fills in the missing original data on the flat objects [20].

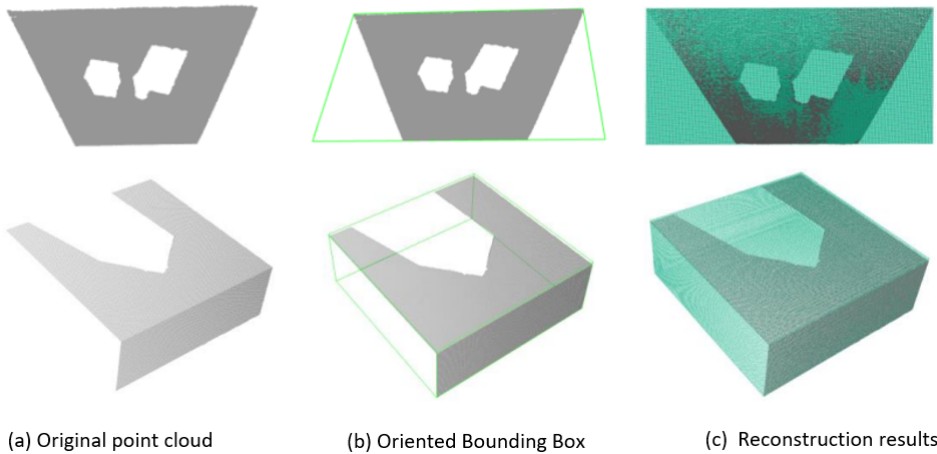

(a) Original point cloud       (b) Oriented Bounding Box       (c) Reconstruction results

**Figure 4.** Reconstructed result diagram of planar objects.

### 3.2.3. Recognition and Modeling of Curved Geometric Objects

Common curved geometric objects include cylinders, cones, and spheres. Due to factors such as the capturing angle and object occlusion, cylinders and cones exist in two forms in actual capturing scenes, namely planar-surfaces combination and curved-surfaces combination. Due to their different manifestations, the logical processing of recognition and parameter extraction for each curved geometric object is also different.

When the segmented object belongs to the planar-surfaces combination type, the relationship between the plane and the adjacent curved surfaces' normal vectors is determined by using the plane as the auxiliary matching surface. If the normal vectors of the two are perpendicular to each other, it is a cylinder, corresponding to objects like cups or other cylindrical objects. If there is no perpendicular relationship between the two normal vectors, and the angle with the normal vector of the adjacent surface remains unchanged, it is a cone.

Due to occlusion or capturing angle, cylinders or cones may also exist with only the curvature of surfaces being captured, in which case, all faces of the object are curved surfaces. In addition to this, spheres are also considered, as they are natural geometric objects consisting only of curved faces. For objects with only curved surfaces, the identification of cylindrical surfaces, conical surfaces, and spherical surfaces is done by analyzing the relative magnitude of the principal curvatures at each point in the object's point cloud data. The values of the principal curvatures can be obtained using the calculation formula, assuming the maximum principal curvature is denoted by $k_1$ and the minimum principal curvature is denoted by $k_2$. The geometric primitive type of the curved surface object can be determined based on the relationship between the extrema of the principal curvatures using the following criteria: (1) Cylinder: $k_1 = 0, k_2 > 0$, and $k_2$ remains unchanged. (2) Cone: $k_1 = 0, k_2 > 0$, and $k_2$ varies. (3) Sphere: $k_1 = k_2$, and both $k_1$ and $k_2$ are positive constants.

After determining the initial geometric primitive type of the segmented object using the aforementioned method, as prior knowledge for the RANSAC algorithm, the corresponding geometric primitives are used for parameter extraction. For spheres, the modeling can be directly accomplished by using the sphere center and radius. However, for cylinders and cones, after obtaining the parameter equations of the cylindrical surface and conical surface, the point cloud of the objects needs to be projected onto the corresponding axis, and the distance between the two farthest projected points is taken as the height of the cylinder and cone.

To visually demonstrate the modeling effectiveness of curved geometric objects, point cloud data of the corresponding surfaces of known geometric parameters and dimensions are generated. The reconstruction result of a single curved surface object in the scene is shown in Figure 5, where Figure 5b represents the oriented bounding box established for

the original point cloud. As can be seen from Figure 5c, for different types of curved surface objects, the reconstructed geometric point cloud fits well with the original point cloud and effectively fills in the missing data based on the obtained 3D model.

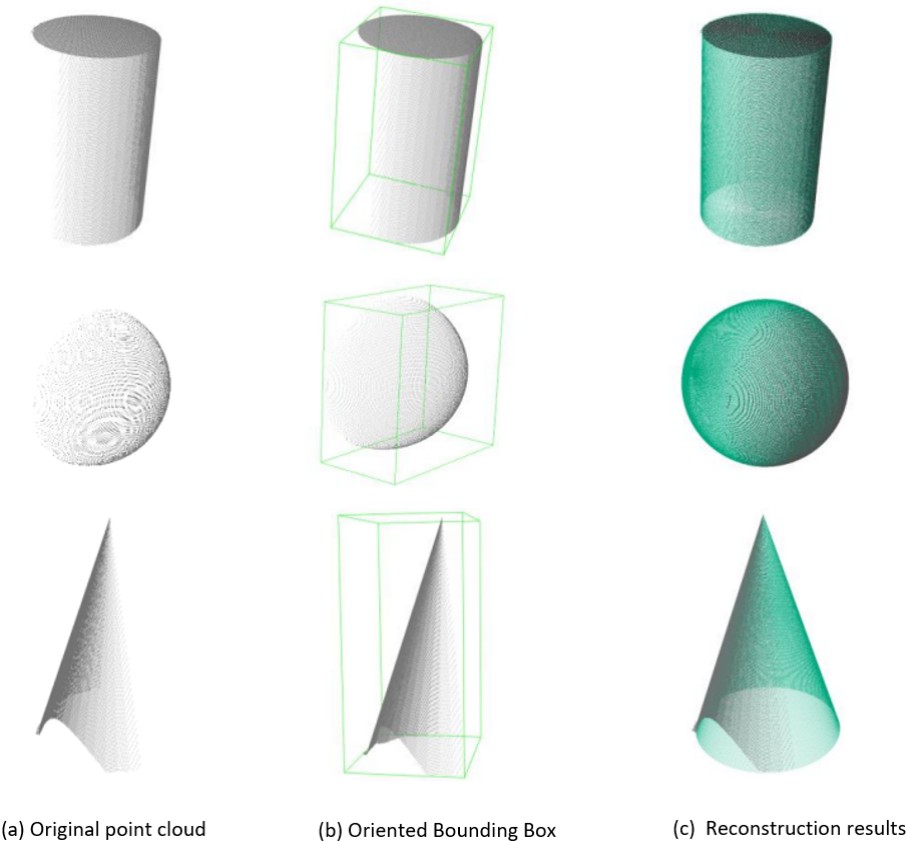

    (a) Original point cloud      (b) Oriented Bounding Box      (c) Reconstruction results

**Figure 5.** Reconstructed result diagram of curved objects.

## 4. Experiments

To verify the accuracy of the algorithm proposed in this paper for recognizing and modeling regular geometric objects, experiments were conducted using synthetic datasets to analyze the errors between the estimated and real values of the dimensions. Moreover, to validate the effectiveness and practicality of the algorithm, experiments were conducted on both public datasets and self-collected datasets, and the reconstruction results of the scenes were visualized. The experiments were conducted in an environment equipped with an Intel i7-10710U CPU @ 1.10 GHz with 16 GB RAM.

### 4.1. Analysis of Experimental Results in Synthetic Dataset

To validate the feasibility of the algorithm proposed in this paper, C++ programming was used to simulate noise-free desktop scene point cloud data generated by a depth camera. Since the accurate dimensions and geometric parameters of different geometric objects cannot be obtained in real-world scenarios, three desktop scenes were synthesized with randomly placed objects of different shapes. The desktop was represented by a plane abstraction, and the desktop objects were represented by a cuboid, cylinder, cone, and sphere. The size of each object in the synthesized scene is known and used to estimate the error between the extracted parameters and the true values. Scene 1 consists of planar objects composed of cuboids with different poses and dimensions. Scene 2 consists of curved objects composed of spheres, cylinders, and cones. Scene 3 is a mixed scene composed of cuboids, spheres, cylinders, and cones.

Figure 6 shows the experimental results of Scene 1, where the input point cloud data contain 372,344 points with an average density of 0.469 mm. From the figure, it can be observed that the plane and four cuboids with different poses and sizes were successfully segmented and labeled with different colors. The segmented objects were further evaluated for their corresponding types and geometric parameters. Based on the extracted parameters, the point cloud data with a specified density were generated as shown in Figure 6c, revealing the recovery of missing data and consistent orientations and positional relationships with the scene point cloud data [21].

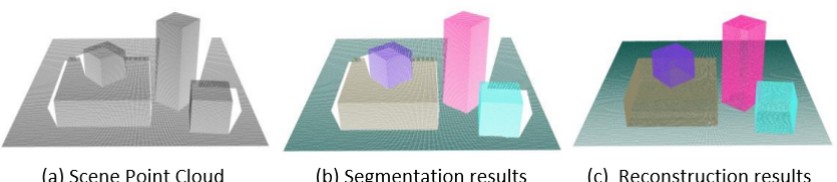

(a) Scene Point Cloud    (b) Segmentation results    (c) Reconstruction results

**Figure 6.** Experimental results of plane object scene.

Figure 7 shows the experimental results of Scene 2. The input point cloud data contain 361,218 points, with an average point cloud density of 0.485 mm. From the figure, it can be observed that the various regular objects on the table are segmented accurately and displayed with different colors. Based on the segmentation results, the types of objects are determined and their parameters are extracted. The reconstruction results, based on the extracted parameters and estimated sizes, are shown in Figure 7c. It can be seen from the figure that our algorithm can effectively recognize and model objects of different sizes, such as spheres, and different orientations, such as cylinders. This algorithm can also complete missing data [21].

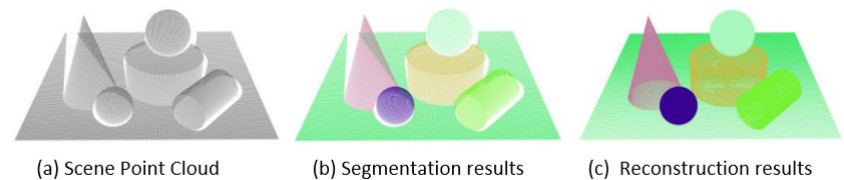

(a) Scene Point Cloud    (b) Segmentation results    (c) Reconstruction results

**Figure 7.** Experimental results of curved object scene.

The experimental results of Scene 3 are shown in Figure 8. The input point cloud data consist of 371,724 points, with an average point cloud density of 0.497 mm. From the graph, it can be observed that both planar and curved objects are completely segmented, and the object reconstruction results are consistent with the positions and orientations of the objects in the original scene point cloud. The experimental results demonstrate that, as the complexity of the scene increases, the algorithm still maintains a high accuracy in object segmentation and geometric parameter extraction.

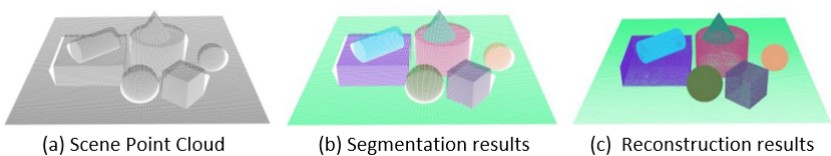

(a) Scene Point Cloud    (b) Segmentation results    (c) Reconstruction results

**Figure 8.** Experimental results of mixed object scene.

To verify the accuracy of the algorithm proposed in this paper for geometric reconstruction, a more complex scenario, referred to as Scenario 3, is taken as an example. The geometric parameters and size values of each object in the scenario are shown in Figure 9. In the figure, the parameter $c$ represents the center point, $\vec{n}$ represents the

normal vector, $\vec{l}$ represents the axial direction, and $p$ represents a point on the axis. These parameters are used to determine the position or orientation of the geometric object in three-dimensional space. The dimensions $L, W$, and $H$ represent length, width, and height, respectively, while $r$ represents the radius, which is used to determine the size of the geometric object in three-dimensional space. Since the sizes of the geometric objects are known when generating synthetic scenes, these known sizes are used as ground truth values to estimate the errors with the sizes estimated by the algorithm in this chapter. From the size errors shown in the figure, it can be observed that the reconstruction accuracy of the algorithm in this chapter is high, with size errors not exceeding 0.5 mm. Compared to the average density of the scene's point cloud [22], the error between the estimated values and the actual values is small, indicating that the algorithm proposed in this paper can effectively model regular geometric objects. Although not reaching zero error, the error is very small.

| object | geometrical parameter (mm) | | true size (mm) | estimation (mm) | error (mm) |
|---|---|---|---|---|---|
| | $c = (-9.030, -43.117, 518.281)$ | | $L = 400.000$ | $L = 399.997$ | 0.003 |
| | $\vec{n} = (-0.000, -0.766, -0.643)$ | | $W = 400.000$ | $W = 400.236$ | 0.236 |
| | | | $L = 50.000$ | $L = 50.005$ | 0.005 |
| | $c = (65.801, 20.251, 403.676)$ | | $W = 50.000$ | $W = 50.461$ | 0.461 |
| | | | $H = 50.000$ | $H = 50.002$ | 0.002 |
| | | | $L = 120.000$ | $L = 120.118$ | 0.118 |
| | $c = (-90.823, -35.929, 478.106)$ | | $W = 120.000$ | $W = 120.007$ | 0.007 |
| | | | $H = 40.000$ | $H = 40.107$ | 0.107 |
| | $p = (23.076, -20.571, 485.669)$ | | $r = 50.000$ | $r = 49.676$ | 0.324 |
| | $\vec{l} = (-0.000, -0.766, -0.643)$ | | $H = 60.000$ | $H = 60.033$ | 0.033 |
| | $p = (89.476, 18.292, 398.252)$ | | $r = 20.000$ | $r = 19.909$ | 0.091 |
| | $\vec{l} = (-0.643, 0.492, -0.587)$ | | $H = 80.000$ | $H = 80.001$ | 0.001 |
| | $p = (-108.941, -111.719, 366.409)$ | | $r = 30.000$ | $r = 29.686$ | 0.314 |
| | $\vec{l} = (-0.000, -0.766, -0.643)$ | | $H = 50.000$ | $H = 49.641$ | 0.359 |
| | $c = (29.504, -99.433, 429.604)$ | | $r = 30.000$ | $r = 29.887$ | 0.113 |
| | $c = (-59.258, 32.188, 381.626)$ | | $r = 25.000$ | $r = 29.921$ | 0.079 |

**Figure 9.** Estimation results of geometric parameters and dimension errors in mixed scene.

To evaluate the efficiency of the algorithm in this paper, Table 1 shows the time required for object segmentation and geometric parameter extraction in different synthetic scenes, including the number of objects on the tabletop. Due to the varying size and complexity of the scene point clouds, the time required for object segmentation and parameter extraction also varies. However, the total time required for a single scene is usually within 3.0 s, which meets the requirements for some real-time operations of indoor robots.

**Table 1.** Running time of the synthetic dataset scenario.

| Dataset Scenes | Points | Number of Objects | Object Segmentation Time (s) | Parameter Extraction Time (s) |
|---|---|---|---|---|
| Scene1 | 372,344 | 5 | 2.372 | 0.171 |
| Scene2 | 361,218 | 6 | 2.256 | 0.223 |
| Scene3 | 371,724 | 8 | 2.425 | 0.282 |

*4.2. Analysis of Experimental Results from Self-Generated Dataset*

To verify the universality of the algorithm proposed in this paper, experiments were conducted using Microsoft Azure Kinect cameras to capture tabletop and ground scenes in the laboratory. The scenes were composed of multiple randomly arranged rule-based objects and everyday items, and the point cloud data contained a significant amount of noise and holes. The experimental results for each scene are shown in Figure 10, which includes the object segmentation results, reconstruction results, and the overlay display of the scene point cloud and reconstruction results. From the figure, it can be seen that for different indoor scenes, the algorithm proposed in this paper can accurately and completely segment various objects. For segmented rule-based objects, point cloud reconstruction is performed based on the extracted geometric parameters. As can be seen from the reconstruction results, regardless of whether it is a planar or curved geometry, the algorithm proposed in this paper is not limited by object pose and size, and can accurately reconstruct point cloud data models that fit the real surface of the objects. The experiments have demonstrated that the proposed algorithm is also applicable to self-captured data, and it exhibits robustness to missing and noisy point cloud data, which has practical application significance for indoor robot perception in unknown environments [23].

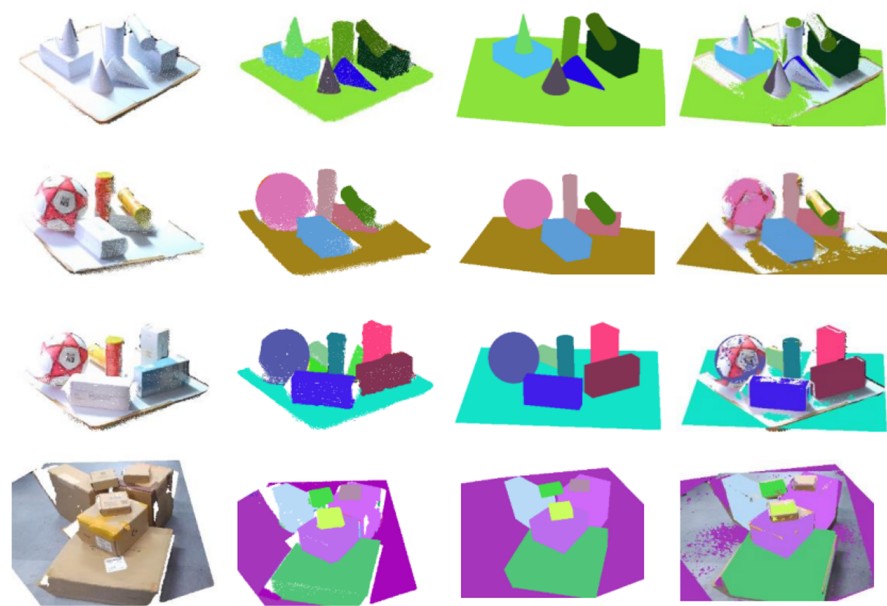

(a) scene point cloud　(b) segmentation results　(c) reconstruction results　(d) overlay results

**Figure 10.** Experimental results of self-collected dataset.

## 5. Conclusions

In this paper, we have presented a geometric modeling algorithm based on known segmentation results, aiming to enrich semantic information for robot understanding in unknown indoor environments. Our algorithm utilizes a flatness criterion to judge surface types within each segmented object. To ensure accurate and stable judgment of flatness, we employ a multi-scale neighborhood approach to calculate curvature and covariance matrix eigenvalues. Furthermore, our algorithm establishes a geometric model based on various combinations of flat surfaces, distinguishing between two major types: flat and

curved surfaces. We apply different analysis and processing methods according to the type of surface present. While our algorithm demonstrates its effectiveness, robustness, and efficiency in accurately modeling regular geometric objects, we acknowledge that it may not be accurately applicable to objects with irregular shapes or combinations of different geometric elements. In addition, in scenarios where low-quality data hinders precise point cloud segmentation or correct classification of objects as planar or curved, the algorithm's performance may be compromised. We conducted experiments using synthetic datasets, public datasets, and self-collected datasets, which confirmed the small error in geometric body size estimation and validated the efficacy and robustness of our proposed algorithm.

**Author Contributions:** Conceptualization, Q.Y.; Methodology, C.W.; Software, T.Y. All authors have read and agreed to the published version of the manuscript.

**Funding:** This research received no external funding.

**Institutional Review Board Statement:** Not applicable.

**Informed Consent Statement:** Not applicable.

**Data Availability Statement:** Not applicable.

**Conflicts of Interest:** The authors declare no conflict of interest.

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
