# Peer review of "Multi-Scale Indoor Scene Geometry Modeling Algorithm Based on Segmentation Results"

_applsci, doi:10.3390/app132111779_

Round 1

Reviewer 1 Report

The authors proposed a new algorithm for modeling regular geometric objects in interiors based on point cloud data. The algorithm is more accurate, efficient, and robust to occlusion and stacking than previous methods. The algorithm is designed for regular geometric objects. For objects with irregular shapes or combinations of different geometric elements, it may not be accurate enough. In the case of low-quality data, it may be difficult to correctly segment the point cloud into individual objects or classify each object as planar or curved. Contribution is clearly described. I recommend editing the conclusion to better reflect the article.

Author Response

  • We pointed out in the conclusion that the action scenario is a regular geometric object, and the performance of the algorithm may be affected in cases where low-quality data hinders accurate point cloud segmentation or the correct classification of objects as planes or surfaces.

Reviewer 2 Report

In this work titled Multi-scale Indoor Scene Geometry Modeling Algorithm Based on Segmentation Results, the authors dealt  with Object Segmentation; Face Recognition; Oriented Bounding Box; Geometric Modeling,.

The topics are presented  into introduction,  Related Work,   Methods,   Experiments , Conclusions,   and  the authors mention  only 23  references

The authors proposed  geometric  modeling algorithm based on known segmentation results.

According to the authors “Experiments were performed on synthetic datasets, public  data sets, and self-collected datasets, and the experimental results showed a small error in geometric body size estimation, verifying the effectiveness and robustness of the algorithm  proposed in this submitted  paper   

The authors mention only few references: I suggest improving the state of the art

Author Response

  • We have updated some references to the latest years and also added some references

Reviewer 3 Report

This manuscript provides a Geometry Modeling Algorithm using multiscale segmentation for indoor scenes.

Furthermore, it presents the results regarding the search neighborhood, Feature Calculation Based on Covariance Matrix, Recognition and modeling of regular geometric shapes,.

Major Comments

1. A quantitative comparison with other studies is not available.

2. The authors mentioned the error details in Figure 9, which are the demerits of the proposed system for not reaching zero error.

3. The procedures are not clearly mentioned when switching from 3.1. Determining the type of planar or curved surface to 3.2 Recognition and modeling of regular geometric shape as shown in Figure 1. To achieve the experimental results of the mixed object scene, as shown in figure 8.

Minor

1. Conclusion: Section 5 is required to improve.

2. Kindly highlight your novelties. 

Author Response

  • We have revised the explanation of the error results, although it did not reach zero error, the error is very small

  • We have improved Section 5 while highlighting our own innovation points
  • We added the steps for comparing and selecting methods and extracting geometric parameters for modeling in Section 3.2.1

Round 2

Reviewer 2 Report

I checked the new version of thev submitrec paper the authors adequately provided the questions formulated by me

 The paper is acceptable as is in its update  format

Reviewer 3 Report

-